# Polish Medical Doctors’ Opinions on Available Resources and Information Campaigns concerning Antibiotics and Antibiotic Resistance, a Cross-Sectional Study

**DOI:** 10.3390/antibiotics11070882

**Published:** 2022-06-30

**Authors:** Olga Maria Rostkowska, Dorota Raczkiewicz, Weronika Knap-Wielgus, Wojciech Stefan Zgliczyński

**Affiliations:** 1Department of Transplantation Medicine, Nephrology and Internal Diseases, Medical University of Warsaw, 02-006 Warsaw, Poland; olga.rostkowska@wum.edu.pl; 2Department of Medical Statistics, School of Public Health, Center of Postgraduate Medical Education, 01-826 Warsaw, Poland; dorota.bartosinska@gmail.com; 3Infant Jesus Clinical Hospital, Medical University of Warsaw, 02-091 Warsaw, Poland; weronika.knap96@gmail.com; 4Department of Lifestyle Medicine, School of Public Health, Center of Postgraduate Medical Education, 01-826 Warsaw, Poland

**Keywords:** antibiotics, antibiotic resistance, antimicrobial agents, physicians’ practice pattern, information seeking, public health, continuing medical education, Poland

## Abstract

*Background:* Antibiotic resistance (ABR) is at the top of global health threats. This paper aims to assess Polish physicians’ readiness to impact ABR through prescribing routines. *Methods:* Surveying Polish physicians participating in specialization courses at the Center for Postgraduate Medical Education in Warsaw, Poland from October 2019 to March 2020. *Results:* Information was obtained from 504 physicians aged 25–59, mean 32.8 ± 5.9 years, mainly women (65%). Most doctors (78%) prescribed antibiotics at least once a week. Physicians indicated clinical practice guidelines as resources most often consulted in the management of infections (90%). However, clinical experience was also declared a powerful resource. In total, 54% of respondents recalled receiving information about the prudent use of antibiotics within 12 months, which partially translated into changing views (56%) and practice (42%). Physicians disagreed that national campaigns provide good promotion of prudent antibiotics use (75%) or that they are effective (61%). Only 40% of doctors were aware of the national campaign promoting responsible antibiotics use, 24% had heard about the European Antibiotic Awareness Day and 20% knew about the World Antimicrobial Awareness Week. *Conclusions:* Prescribers most often rely on clinical practice guidelines and their own experience as resources for antibiotics use. Doctors’ awareness of available resources and information campaigns concerning antibiotics and antibiotic resistance should be improved.

## 1. Introduction

While in the last 2.5 years, much of the public and medical attention was driven towards COVID-19, antibiotic resistance (ABR) has remained the proverbial sword of Damocles in global health. According to Bassetti et al., by 2050 the mortality rates due to ABR will exceed those caused by cancer (10 mln vs. 8.2 mln), if our commitment does not match the scale of the problem [1]. There are different tools to mitigate the threat of ABR. Many of them have already seen varying levels of implementation. Those strategies include policy adaptation, stronger surveillance with scrupulous epidemiological tracking, quicker diagnostics, improved application of treatment guidelines as well as research into novel antimicrobials and ways to enhance patients’ immunity, to name a few [2,3]. At the same time, healthcare workers’ commitment alone plays a fundamental role in antimicrobial stewardship, i.e., ensuring prudent distribution, administration and use of antibiotics [4,5,6].

As in many public health domains, prevention is of colossal importance in this matter. Mass education on ABR, both structural and informal, e.g., via social media, is necessary [2,7]. Nonetheless, regardless of technological advancements, direct communication in healthcare is fundamental, as we learnt during the recent COVID-19 outbreak or in any vaccination advocacy effort [8,9,10,11]. Medical doctors are a melting pot when it comes to spreading awareness and promoting responsible application of antibiotics, as they decide upon the prescription and communicate regarding their therapeutic choices with a patient [12,13]. In order to achieve that, medical doctors and dentists need to match solid knowledge on antimicrobials with empathic resonance towards patients. This sometimes poses the difficulty of juggling professional assertiveness with respect for the patient’s autonomy when defining “pros” and “cons” of prescribing and using antibiotics [14,15,16]. However, the stakes are high. Preserving the function of available antimicrobials as long as possible is currently seen through the lens of medical ethics [17,18]. Furthermore, protection of antibiotics as precious resources has become a problem beyond human healthcare. Since antibiotics are globally used in animal husbandry to an even greater extent than in human medicine, the ABR crisis blends with agriculture under the One Health approach [19,20,21]. Both human and veterinary medicine are combined vessels.

Based on the OECD and ECDC data from 2015, Poland recorded 2218 deaths from infections caused by ABR bacteria, making it the fourth-highest number in the EU/EEA, with a total of 33,110 for all countries [22]. According to the data on Poland from the ECDC Surveillance Atlas on antimicrobial resistance, the percentage of *K. pneumoniae* isolates not susceptible to carbapenems went up from 0.5% in 2015 to 8% in 2020. For E. coli, combined resistance to third-generation cephalosporins, fluoroquinolones and aminoglycosides in the same time period has increased from 6.1% to 9.4% [23]. As reported by Wojkowska-Mach et al., the total use of systemic antibiotics with focus on β-lactamase-sensitive penicillins has undergone a significant increase in the years 2007–2016 (22.2 to 23.9 DIDs and from 0.8 to 1.3%, respectively) [24]. In the analysis of antibiograms from a sample of Polish transplant patients, a significant decrease in the susceptibility of E. coli to amoxicillin/clavulanic acid (63% vs. 40.0%) and ciprofloxacin (100% to 40.0%) was noted in the years 2011–2018 [25]. Overprescribing and overconsumption of antimicrobials in Poland, which contribute to the growth of ABR burden, are critical matters. In a 2017 paper by Mazińska and Hryniewicz, most physicians (91%) agreed that antimicrobial resistance is a problem [26]. As reported in survey results performed by Zgliczyńki et al., over 98% of interviewed physicians see the connection between prescribing antibiotics and ABR [27]. The problem of antibiotics-prescribing patterns among Polish physicians has been explored to a lesser extent thus far.

Each year, November 18th marks the European Antibiotic Awareness Day (EAAD) as part of the World Antibiotic Awareness Week (WAAW) [28]. During this period, a number of online and offline initiatives take place which aim to bring the challenge of antibiotic preservation, reasonable use of these medicines and bacterial infection control into the public spotlight. Events organized include seminars, expert meetings, social media campaigns and release of informative visuals for the public as well as targeting healthcare workers. Poland joins these efforts through the National Programme for the Protection of Antibiotics (NPPA) established by the Ministry of Health and coordinated by the National Institute of Medicines [29]. Since 2008, actions included releasing materials for patients, doctors and hospital administration in Poland and keeping records of the public campaigns on ABR across the country. The main focus of this paper is to gauge the opinion of doctors in Poland on available resources and information campaigns on antibiotics and antibiotic resistance. This is the second part of a research study on the management of antibiotics by Polish medical doctors carried out by the School of Public Health, Center of Postgraduate Medical Education (CMKP) in Warsaw, Poland [27]. The whole research project was based on a survey and a report released by the European Centre of Infection Control and Prevention (ECDC) on knowledge, attitudes and behaviors regarding antibiotics, antibiotic use and antibiotic resistance among healthcare workers across 30 EU/EEA countries [30,31]. Our data complete and sometimes challenge information collected by the ECDC with responses from our independent sample of Polish physicians. Moreover, presented outcomes can inform institutions in charge of continuous professional development of doctors in Poland about some indispensable action points, if we want to address the issue of ABR in human medicine.

## 2. Results

### 2.1. Characteristics of the Study Group

A total of 504 questionnaires that were correctly completed were included in the study. The response rate was 81%.

The data were obtained from medical doctors aged 25–59, mean age 32.8 ± 5.9 years, mainly women (65%), mostly living in cities with more than 500,000 inhabitants (54%). Figure 1 presents the demographic characteristics of the study group. The same study group’s characteristics were presented in our previous research on knowledge and practice of Polish medical doctors on antibiotics prescribing and antibiotic resistance [27].

The surveyed medical doctors had work experience as medical doctors from 1 to 39 years, 5.1 ± 5.8 years, on average. Figure 2 presents the professional characteristics of the study group and frequency of prescribing or administering antibiotics.

### 2.2. Resources Used by Medical Doctors in the Management of Infections

The resources consulted by the participants in the management of infections are presented in Table 1.

Regarding resources used in the management of infections, females more commonly indicated clinical practice guidelines (92.1% of women vs. 84.7% of men, *p* = 0.011) and continuing education training courses than men (48.0% of women vs. 37.7% of men, *p* = 0.027).

Clinical practice guidelines were more frequently indicated by doctors working in a clinical hospital rather than in other places (93.0% vs. 87.6%, *p* = 0.049), by those living in cities rather than in towns or villages (91.6% vs. 85.4%, *p* = 0.037), by younger respondents (92.0% aged 20–39 years vs. 71.7% aged 40+ years, *p* < 0.001) and by those with lower job seniority (92.0% working for 1–7 years vs. 78.2% working for 8+ years, *p* = 0.001).

Documentation from pharmaceutical companies was more frequently consulted by doctors with surgical specializations (15.2% vs. 7.7%, *p* = 0.033), by those working in other places than in a clinical hospital (10.7% vs. 5.5%, *p* = 0.040), by those living in towns or villages (14.6% vs. 6.6%, *p* = 0.005) and by older respondents (18.9% aged 40+ years vs. 6.7% aged 23–39 years and 8.2% aged 30–39 years, *p* = 0.022).

The frequency of using other resources in the management of infections did not correlate with the characteristics of the respondents. Frequency of using all the resources listed in Table 1 did not correlate with the status of a specialization (completed or not), or with the frequency of prescribing/administering antibiotics (*p* > 0.05).

Table 2 presents numbers and percentages of the respondents who received information on avoiding unnecessary prescribing, administering or dispensing of antibiotics; as well as of those reporting that the information contributed to changing their views or practice. Out of the total of 504 respondents surveyed, 270 respondents (53%) said they received information about avoiding unnecessary prescribing/administering/dispensing of antibiotics during the previous year. Out of those, 56% changed their views and 42% changed their practice on prescribing/administering/dispensing antibiotics.

Receiving information about avoiding unnecessary prescribing or administering or dispensing of antibiotics correlated with the frequency of prescribing/administering antibiotics (*p* = 0.007, Figure 3). Receiving information about avoiding unnecessary prescribing or administering or dispensing of antibiotics and changing views or practice based on this information did not correlate with the characteristics of the respondents (*p* > 0.05). Moreover, changing views or practice based on this information did not correlate with the frequency of prescribing/administering antibiotics (*p* > 0.05).

Regarding sources of information about avoiding unnecessary prescribing, administering or dispensing of antibiotics and the sources of information which had the most influence on changing the doctor’ views, respondents most often mentioned the same three sources: published guidelines; colleague or peer; training—group (Table 3).

Out of the 112 respondents (from Table 2) who did not change their practice based on the information they received, 96 (85.7%) said this was because they were already following the principles of the message. None of the respondents chose the following answers: “forgot about the message”, “do not think the message is important”, “no control over it”, “information was not relevant to my practice”.

Regarding the sources of information about avoiding unnecessary prescribing/administering/dispensing of antibiotics listed in Table 3, social media were more often indicated by doctors working in places other than a clinical hospital (14.2% vs. 6.1% of those working in a clinical hospital, *p* = 0.041) and by younger respondents (17.9% aged 23–29 years vs. 10.1% aged 30–39 years and none of the respondents aged 40+ years, *p* = 0.032). A colleague or peer was indicated as a source of information about avoiding unnecessary prescribing/administering/dispensing of antibiotics more often by the younger respondents (65.5% aged 23–29 years and 57.2% aged 30–39 vs. 24.0% aged 40+, *p* = 0.001) and by those with lower job seniority (65.4% working for 1–3 years vs. 47.1% working for 4–7 years and 40.4% working for 8+ years, *p* = 0.002). Training (any type) was indicated as a source of information more often by respondents with higher job seniority (19.1% working 1–3 years vs. 23.5% working 4–7 years and 36.2% working 8+ years, *p* = 0.050).

Indication of a colleague or peer as a source of information about avoiding unnecessary prescribing/administering/dispensing of antibiotics correlated with the frequency of prescribing/administering antibiotics (*p* = 0.006, Figure 4).

The frequency of using other sources of information about avoiding unnecessary prescribing/administering/dispensing of antibiotics did not correlate with other characteristics of the respondents. Moreover, the frequency of all the sources listed in Table 3 did not correlate with gender, place of residence or whether specialization was completed or not (*p* > 0.05).

Regarding sources of information which had the most influence on changing the respondent’s views, listed in Table 3, an employer was indicated only by the respondents with non-surgical specialization (12.7% with non-surgical vs. none with surgical, *p* = 0.046) and only by the respondents with the lowest and the highest job seniority (15.7% working for 1–3 years and 3.5% working for 8+ years vs. none working for 4–7 years, *p* = 0.019). An employer was indicated more often by the youngest and the oldest respondents (23.1% aged 23–29 and 11.1% aged 40+ vs. 5.4% aged 30–39, *p* = 0.011). Public policy was indicated only by respondents working in a clinical hospital (5.6% vs. none working in other places, *p* = 0.020). Individual or specialized training was indicated more often by doctors with the highest job seniority (38.9% working for 8+ years vs. 10.3% working for 1–3 years and 16.1% working for 4–7 years, *p* = 0.026).

The frequency of other sources of information which had the most influence on changing the respondents’ views did not correlate with characteristics of the respondents. The frequency of all the sources listed in Table 3 did not correlate with gender, place of residence, specialization status (completed or not) and with the frequency of prescribing antibiotics.

### 2.3. Campaigns and Training

Table 4 presents information channels on ABR categorized by their perceived effectiveness.

Regarding levels considered most effective in knowledge dissemination on ABR/antibiotics, females more frequently indicated all levels (65.8% vs. 47.1%, *p* < 0.001), while less often indicated individual in public (12.5% vs. 21.2%, *p* = 0.012) and European/global level (5.5% vs. 14.7%, *p* = 0.001). Patient-healthcare worker communication was more often indicated by respondents with non-surgical specialization (24.9% vs. 13.9%, *p* = 0.035). Respondents with the lowest job seniority indicated all levels more often (65.2% working for 1–3 years vs. 52.9% working for 4–7 years and 56.3% working for 8+ years, *p* = 0.050) and less often individual in public (10.7% working for 1–3 years vs. 24.0% working for 4–7 years and 19.5% working for 8+ years, *p* = 0.002). Messages related to environment or animal health were more frequently indicated by older respondents (17.0% aged 40+ vs. 5.3% aged 23–29 and 6.5% aged 30–39, *p* = 0.035) and those with higher job seniority (12.6% working for 8+ years vs. 4.8% working for 1–3 years and 8.7% working for 4–7 years, *p* = 0.035).

The choice of other communication levels did not correlate with the characteristics of respondents. Moreover, the choice of all the levels listed in Table 4 did not correlate with the place of residence and work, with specialization status or the frequency of prescribing antibiotics.

Table 5 presents the initiatives undertaken in Poland focusing on antibiotic awareness and ABR.

Out of the initiatives listed in Table 5, conferences or events focused on tackling antibiotic resistance were more commonly indicated by the older respondents (58.5% aged 40+ years vs. 36.7% aged 23–29 and 40.1% aged 30–39, *p* = 0.019) and with higher job seniority (57.5% working for 8+ years vs. 37.6% working for 1–3 years and 37.5% working for 4–7 years, *p* = 0.003).

National campaigns were more frequently indicated by men (18.8% vs. 8.6%, *p* = 0.001) and by the older respondents (22.6% aged 40+ years vs. 7.3% aged 23–29 and 12.3% aged 30–39, *p* = 0.012). National or regional posters or leaflets on antibiotic awareness were more commonly indicated by the respondents working in a clinical hospital (39.3%) than those working in other places (28.5%), (*p* = 0.012) and by the respondents living in cities (35.6% vs. 26.5%, *p* = 0.048).

The knowledge of other initiatives undertaken in Poland focusing on antibiotic awareness and antibiotic resistance did not correlate with the characteristics of the respondents. Moreover, knowledge of all the initiatives listed in Table 5 did not correlate with specialization status, the type of specialization or the frequency of prescribing antibiotics.

Only 11.5% of the respondents agreed or strongly agreed that there had been good promotion of prudent antibiotic use and information about antibiotic resistance in Poland, and only 20.5% agreed or strongly agreed that national campaigns were effective in reducing unnecessary antibiotics use and controlling ABR (Table 6).

Analyzing the results from Table 6 in a 1–5 point scale, we obtained the average of 2.1 ± 1.0, which means the respondents disagreed that there had been good promotion of prudent antibiotics use and information about antibiotic resistance in Poland, and the average of 2.4 ± 1.1, which means the respondents’ answers were between “disagree” and “undecided” that national campaigns were effective in reducing unnecessary antibiotic use and controlling antibiotic resistance.

When we correlated the responses in a 5-point scale for two of the above-mentioned questions with the respondents’ characteristics and the frequency of prescribing antibiotics, we obtained only one significant correlation. The respondents with higher job seniority agreed more that national campaigns were effective in reducing unnecessary antibiotics use and controlling antibiotics resistance (average 2.7 ± 1.1) compared to respondents with lower job seniority (average 2.4 ± 1.1 for working for 1–3 years and 2.3 ± 1.1 for working for 4–7 years) (*p* = 0.020).

### 2.4. Awareness of the National Action Plan on Antimicrobial Resistance, European Antibiotic Awareness Day and World Antibiotic Awareness Week

In total, 40.1% of the respondents were aware that Poland has a national action plan on antimicrobial resistance (the National Plan for the Protection of Antibiotics), 23.8% had heard of the EAAD and 19.8% had heard of the WAAW (Table 7).

When we correlated the responses to the three questions from Table 7 with the respondents’ characteristics and the frequency of prescribing antibiotics, we obtained only significant correlation with a place of residence (Figure 5).

Only 4.3% of the respondents believed that the EAAD had been effective or very effective in raising awareness about the prudent use of antibiotics and ABR, and only 5.2% believed that the WAAW had been effective or very effective (Table 8).

Analyzing the results from Table 8 in a 1–5 point scale, we obtained the average of 2.7 ± 0.7 for both EAAD and WAAW, which means close to undecided. When we correlated the responses in a 5-point scale for two of the above-mentioned questions with the respondents’ characteristics and the frequency of prescribing antibiotics, we did not obtain any statistically significant results.

## 3. Discussion

To the author’s knowledge, this is the only published and by far the most recent study of Polish physicians’ opinions on available resources and information campaigns on antibiotics and antibiotic resistance. It is a continuation of a study by Zgliczyński et al. on knowledge and practice in antibiotic management among Polish prescribers published in 2022 [27].

### 3.1. Resources Used by Medical Doctors in Management of Infections

Antibiotic resistance has been a problem known for as long as antibiotics themselves and was already addressed by Sir Alexander Fleming in his Nobel Prize Lecture in 1945 [32,33]. The phenomenon of excessive prescribing of antibiotics and their inappropriate use is in part the result of insufficient knowledge of the doctors prescribing them as well as suboptimal ABR communication engagement with patients [34,35,36]. The issue of ABR has not been sufficiently acknowledged by society and it has often been ignored by physicians, who go against the scientifically sound recommendations for prescribers—intentionally or not [37,38,39,40]. That is why various methods meant to improve physicians’ knowledge and decision-making patterns, considering indications for antibiotic therapy, its goals, benefits and side effects, have been so important [6,41,42,43]. Already during their studies, young adepts of medicine across the world have the opportunity to familiarize themselves with specific recommendations for the treatment of most common infections, such as those affecting the respiratory system or genitourinary tract [44,45,46,47]. Clinical practice guidelines certainly facilitate making decisions on the choice of a specific antibiotic regimen. In our study, when asked about the most frequently used resources for anti-infective therapy, 89.9% of the respondents indicated clinical practice guidelines. It is important not only to continuously promote such tools but also to advocate for open-access and easy-to-use formats of guidelines [48]. It is commendable that aside from international or national guidelines, local and site-specific recommendations are being released, e.g., for primary care physicians [49,50]. Local guidelines are especially important in departments where antibiotics use is exceptionally common, thus where ABR rates can be elevated, for instance, in transplant clinics [25]. Over time, however, it is the clinical experience that largely influences the management of a patient. As many as 73.2% of the physicians participating in our study claimed that their actions were based on the clinical competencies they had gained thus far. Considering that more and more doctors specialize in narrow fields of medicine, consultations with specialists in pharmacology or microbiology have rightfully become a standard in many practices [51,52,53]. In fact, American and German scientific societies recommend consulting multidisciplinary teams, including an infectious diseases physician, pharmacist and epidemiologist, when making decisions regarding treatment with antibiotics [53,54]. In our study group, more than a third of respondents consult other healthcare professionals before making a final decision as to whether and how to use antibiotics. At the same time, medical documentation, social media and materials presented by representatives of pharmaceutical companies were least mentioned. Based on the available Swedish and German studies, interactions between physicians and pharmaceutical representatives promoting specific medicines can have negative consequences, such as unjustified prescriptions, elevated financial costs and, above all, lower quality of treatment [55,56]. When making decisions about antibiotic therapy, it is important for physicians to pay attention not only to the analysis of data, but also to the background of authors and sponsors behind given resources.

In our sample, 54% of respondents received information within the previous 12 months on how to avoid unnecessary prescribing of antibiotics. Of those, 56% agreed that this information had impacted the way they use antibiotics in practice. In the ECDC study on healthcare workers (HCW) in Europe, from those who received information on antibiotics within the last year, 58% changed their view while 42% changed their practice [30]. Respondents from a mirror study performed in Jordan displayed much higher compliance with the information received on better use of antimicrobials, i.e., 94% declared a tangible change in their practice upon receiving dedicated information [57]. It poses a research question as to why information resources fail to change to a greater extent the perspectives of European doctors on how to be a more responsible antibiotics prescriber.

Regarding sources of information about avoiding unnecessary antibiotics prescribing and the sources of information which had the most influence on changing the doctors’ views, physicians in our sample favored clinical practice guidelines (67.0%), colleagues or peers (56.7%) or training—conference groups (54.8%). Our findings resonate with data obtained from the Polish HCW who participated in the ECDC study, who indicated that aside from clinical guidelines, group training (e.g., at conferences) and individual training (e.g., studying) were preferred sources of information on prescribing antibiotics. Overall, according to the ECDC report, clinical guidelines were chosen by all healthcare workers from all countries over all other resources (66%) [30,31]. This mirrors the results of a study by Sychareun et al. on using antibiotics in pregnant women and around birth in Laos where national guidelines and the World Health Organization guidelines were the most commonly consulted sources [58]. On the other hand, consulting guidelines as a preferred resource do not always translate to compliance. According to a study by Hohmann et al. on prophylaxis in surgery in Germany, which included over 6000 patients’ data, dedicated guidelines were followed in approximately 71% of cases [59]. In a study from Aarhus University Hospital in Denmark (data 2001–2014), 37% of patients with acute tonsillitis were treated against the available antibiotic guidelines [60]. Similarly, findings delivered by Hek et al. from the Dutch primary care setting demonstrate that in the treatment of respiratory tract infections (e.g., acute bronchitis, tonsillitis, strep throat), unnecessary prescribing of antibiotics can be as high as 50% [61]. A 2016 report from the Centers for Disease Control and Prevention on ambulatory care in the US suggests that approximately every third antibiotic prescribed was not in line with available guidelines [62]. The problem also exists in pediatrics. According to a study by Ivanovska et al., physicians in primary care in The Netherlands prescribed antibiotics to children with bronchitis in 40% of cases, which fairly exceeds the guidelines threshold [63]. All in all, most studies across different settings and periods demonstrate suboptimal compliance of doctors with available guidelines, despite physicians allegedly considering them a valuable source of information regarding antibiotic treatment. Examined reasons for non-compliant administration of antibiotics can be the fear of deterioration of the patient’s condition, demand for a “quick fix”, being unsure of the etiology and pressure from patients [64,65,66]. Ways to address those root causes should be further explored as this could help in making a large step towards better doctors’ adherence to prescription standards across various medical environments.

### 3.2. Opinion on Social Campaigns and Other Methods of Raising Awareness on Antibiotics

Raising social awareness on ABR is essential in curbing this threatening phenomenon. Respondents of our survey declared that taking multidirectional action is the key (59%), i.e., combining global and regional campaigns and media activities with better individual communication with patients in a healthcare setting. Physicians in our group see themselves as the second most important resource for society, with 27% positioning individual impact (i.e., from the prescribing doctor) as the most influential form of advocacy on antibiotics. In fact, doctors’ educational potential for society should not be taken lightly. According to a report by the Standing Committee of European Doctors, over 70% of European doctors declare that they are in a good position to provide information on responsible use of antimicrobials [67]. Moreover, doctors are often seen as those from whom information on antibiotics is expected and they are looked upon as an authority in the field [2,68]. At the same time, in a study of 139 junior doctors working in France and Scotland, 95% of respondents agreed that ABR is a national challenge, while only 63% see it as a daily practice problem [69]. In a study of 695 Caribbean physicians by Nicholson et al., doctors were much worried about ABR in the global aspect (82%), yet less concerned nationally (73%) and even less in an individual patient-doctor context (53%) [70]. According to a study of 215 doctors in India, ABR was seen as a global (66%) and a national (68%) problem, but was considered significantly less important in a hospital setting from an individual standpoint (31%) [71]. If replicated in larger studies, this could indicate relevant educational directions for CME/CPD institutions, since doctors should be encouraged to view ABR as a problem directly affecting their everyday work. This is fundamental, considering doctors’ role in the forefront of expanding social awareness on responsible antibiotics use.

### 3.3. National Action Plan on Antimicrobial Resistance, EAAD and WAAW

Literature review of 22 evaluations of public campaigns targeting patients and/or physicians, such as EAAD and WAAW, demonstrates a probable effect in improving antibiotics use, even if the individual immersion is moderate [72]. Bhattacharya et al. explored challenges in keeping UK-based healthcare professionals engaged in public health work on ABR and prudent use of antibiotics, specifically in the context of EAAD and WAAW. According to the authors, running an individual pledge-based engagement system (becoming an “Antibiotic Guardian”), making it public (e.g., via social media) and building a community around it is an example of a promising strategy [73]. Healthcare workers who commit to making changes in their daily actions around antibiotics, e.g., using guided prescribing, implementing infection prevention or offering education, can make a substantial contribution to antimicrobial stewardship and help correct patients’ behaviors. As Lokhorst and colleagues described, personal commitment to a cause in advocacy is crucial [74]. The benefits of having HCW actively engaged in antibiotics advocacy cannot be overestimated. Pinder et al. found that as much as public health campaigns are good at spreading information on a large scale, it is HCW who make a greater impact on the patient’s actions regarding antibiotics use [75]. It is also important to see the large role of community pharmacists in disseminating knowledge on antibiotics and ABR [76]. Most authors agree that a mixed method approach, i.e., making the public aware via social campaigns whilst keeping professionals educated, engaged and eloquent on the cause of antibiotics can bring us closer to the desired public health outcomes of slowing down ABR.

Physicians in our study disagreed or strongly disagreed that available national campaigns provide good promotion of sensible antibiotics use and information about antibiotics resistance (75%), or that they are effective in reducing unnecessary use of those medicines and controlling ABR (61%). In Poland, one of the awareness raising campaigns is EAAD as a part of WAAW, both of which are powered by the National Programme for the Protection of Antibiotics (NPPA) [29]. More information on each campaign can be found on a Polish site: http://antybiotyki.edu.pl/ (accessed on 26 June 2022). Only 40% of doctors we interviewed were aware of the NPPA (53% were uncertain of their answer), 24% had heard about EAAD and 20% knew about WAAW. In the ECDC study of HCW, analysis of the Polish respondents demonstrated that 35% were aware of NPPA (ca. 60% were uncertain of their answer), 42% had heard about EAAD and 33% had heard of WAAW [31]. In the study by Barchitta et al., the Italian action plan on antibiotics use was known to 32% of sampled doctors, whereas 28% and 24% of physicians were familiar with EAAD and WAAW, respectively, which echoes our findings rather than those from the ECDC report [4]. When asked about the effectiveness of EAAD and WAAW, of those included in our research, only 4.3% and 5.2% were of the opinion that EAAD and WAAW are effective in raising awareness of antibiotics use, respectively. The majority remained undecided (69% for both). In the Polish sample of HCW from the ECDC report, approximately 17% believed in the effectiveness of the EAAD and 14% considered WAAW effective. For the entire HCW group in the ECDC study, the ratios were higher, i.e., 29% thought EAAD was effective and 23% declared so for WAAW [31]. We believe those discrepancies can stem from potential bias of individuals who filled the survey for ECDC as they could have been more aware of the public health activities those agencies deliver, since the survey was launched through the channels of ECDC and Public Health England. Our sample represents a group of doctors selected more randomly who are not directly linked with any public health agency. Interestingly, our study sample consisted of many junior doctors who regularly attend different training, courses and seminars in their specialty program, including public health seminars. Yet, their awareness and belief in the effectiveness of EAAD/WAAW/NPPA was rather low and so was expected engagement in such campaigns. Moreover, a higher percentage of doctors living in towns or villages had heard about EAAD compared to those living in cities (30.4% vs. 21.8%, *p* = 0.026). This could be explained by the scarcity of educational materials in smaller locations and stronger reliance on mass on-line campaigns.

On the other hand, it is puzzling that doctors are not aware of resources about antibiotics and ABR which target patients. The EAAD and WAAW resources together with other original ECDC materials have been available on-line, in Polish, free of charge and presented in various handy formats (infographics, posters, animations, etc.) [28]. Doctors in Poland complain about not having adequate tools and patient-friendly resources at hand to educate their patients on ABR [27]. Perhaps physicians should be familiarized more directly with the selection of materials on WAAW and EAAD through the ECDC sites or the NPPA as part of their in-practice training.

### 3.4. Limitations

According to Statistics Poland (Główny Urząd Statystyczny, GUS), by the end of 2019 there were over 150,000 practicing doctors in the country. Doctors in our study represent only a fraction of the entire population of physicians and mostly they were junior doctors (79% declared 1–7 years of practice). However, our group represents doctors of all basic specialties, from different regions of Poland, installed in various healthcare institutions in the country.

Comparisons of our results with the outcomes of the ECDC study and Italian study by Barchitta et al. [4] should be critically adjusted, considering the discrepancy in data collecting formats, i.e., paper survey vs. online questionnaire. Nonetheless, in the case of paper-based questionnaires filled out in controlled environments such as registration-only specialty courses, we can be certain of the participants’ background. The ECDC study was delivered through social media, email and was available via the agency website, which can pose a challenge in verifying the respondents’ professional category.

The data were collected in 2019 and 2020 and published in 2022. Although the information is from over two years ago, we believe its validity has not been affected. According to our monitoring, no breakthrough programs or information campaigns targeting doctors in Poland were launched throughout this time, partially because of the COVID-19 pandemic capturing most of the public health channels and medical care efforts as well as financial resources over the past several months. In 2022, we hope to see educational and institutional outlets in Poland becoming more generous with information on both ABR and responsible use of antibiotics directed at the public as well as addressing physicians.

## 4. Materials and Methods

### 4.1. Study Group

This study is the second part of a research project dedicated to knowledge, attitudes and behavior related to antibiotics, antibiotic use and antibiotic resistance among Polish medical doctors. The first part of the project was aimed at evaluating knowledge and practice of Polish doctors on antibiotics prescribing and antimicrobial resistance [27].

The study group consisted of doctors participating in specialization courses at the School of Public Health, Center for Postgraduate Medical Education in Warsaw (CMKP), Poland from October 2019 to March 2020.

### 4.2. Survey Questionnaire

#### 4.2.1. Construction of the Questionnaire

The questionnaire included characteristics of the respondents (gender, age, place of residence) and professional characteristics (place of work, type and status of specialization). Questions were divided into three types: multiple-choice, true or false, and agree or disagree with a 5-point Likert scale. The total number of questions was 31. The questionnaire used in this study can be made available in Polish upon request to the corresponding author.

The questionnaire was piloted on a sample of 15 respondents working as medical professionals at the School of Public Health, Centre of Postgraduate Medical Education. As a result of the pilot study, some questions have been modified.

#### 4.2.2. Dissemination Process and Data Collection

The research tool was a self-administrated questionnaire prepared in Polish on the basis of the validated tool used by the ECDC and described in the technical report and the following publication [30,31]. The questionnaire was delivered to participants in paper form by a researcher at the beginning of each course conducted by CMKP. The researcher informed participants about the survey purpose and technical details. The participants could decide whether to take part in the survey or not. Anonymous questionnaires were collected at the end of every course (participants had 3–5 h to return a completed form). Although participation in the study was anonymous and voluntary, each surveyed participant upon returning the questionnaire gave us verbal consent to use the information provided.

Upon completion of the data-collecting step, all information was translated from paper forms filled individually by respondents into electronic format using Microsoft Excel. This process was performed by the office workers of the School of Public Health, CMKP. Further analyses were performed using a digital database.

### 4.3. Statistical Methods

STATISTICA 13.1 software (STATSOFT, Kraków, Poland) was used in statistical analyses. The mean (M) and standard deviation (SD) were estimated for numerical variables, as well as absolute numbers (*n*) and percentage (%) of the occurrence of items for categorical variables.

Pearson’s chi-square test or Fisher’s exact test were used to compare categorical responses to questions between men and women, specialization completed and ongoing, surgical and non-surgical specialization, clinical hospital and other places of work, living in cities and towns or villages. The latter test was used only if at least the expected count was smaller than 5. Student’s t-test was used to compare one variable in ordinal scale between two groups (men vs. women, specialization completed vs. ongoing, surgical vs. non-surgical specialization, clinical hospital vs. other places of work, living in cities vs. towns or villages). Analysis of variance F test was used to compare one variable in ordinal scale between more than two categories of a categorical variable.

The significance level was assumed to be *p* < 0.05 in all the statistical tests.

### 4.4. Ethical Statement

The presented study is not a medical experiment, so according to the Act of 5 December 1996 on the professions of physicians and dentists (Journal of Laws of 2021, item 790 as amended), no ethical consent is required. The study is in line with the Institutional Ethics Committee and the Helsinki Declaration (1964). The agreement to perform the study was granted from the Dean of the School of Public Health, CMKP.

## 5. Conclusions

For Polish medical doctors in our sample, clinical experience gains more value in the decision-making processes on antibiotics use and infection management along with years of practice. Doctors are receptive to new information on antibiotics, although knowledge retention should be improved. Physicians see themselves as an important resource for patients in terms of delivering information on antibiotics. Awareness of social campaigns such as EAAD and WAAW among doctors is moderate, whereas belief in their effectiveness is low. Awareness of the National Programme for the Protection of Antibiotics is very low.

### Recommendations and Future Directions

Based on our findings, we would like to make recommendations for further research and action points for continuous professional development for physicians in Poland:<1>Clinical practice guidelines are highly quoted as a preferred resource but Polish doctors’ adherence may vary. This topic should be explored further in research.<2>It is necessary to encourage doctors to regularly update their knowledge on the innovations in antibiotic therapy (specific to a field of medicine). Methods of knowledge dissemination and awareness raising should be adjusted to the needs of Polish prescribers (e.g., leaflets, workshops, quizzes, higher CME/CPD scoring system).<3>Doctors should be better acquainted with educational opportunities such as World Antimicrobial Awareness Week, European Antibiotic Awareness Day and National Programme for the Protection of Antibiotics. Ways to promote those campaigns more effectively should be explored in the Polish context (e.g., bringing it up in undergraduate medical education).<4>Keeping physicians educated, engaged and eloquent about antibiotics and antibiotic resistance should be a priority for public health agencies and training institutions in Poland.

## Figures and Tables

**Figure 1 antibiotics-11-00882-f001:**
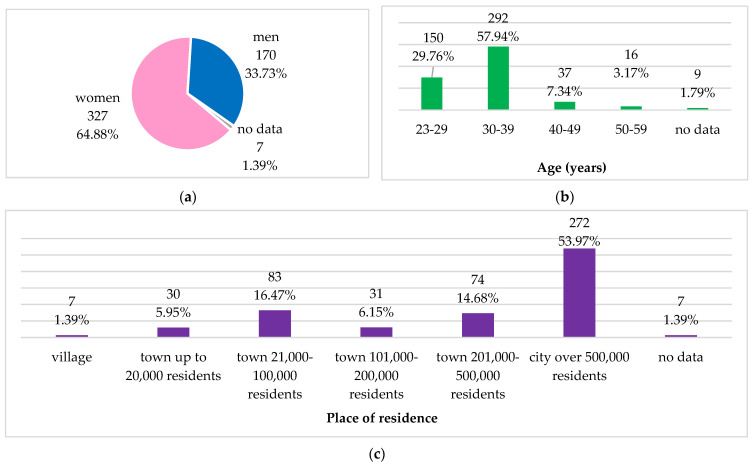
Demographic characteristics of the study group: (**a**) gender; (**b**) age groups; (**c**) place of residence.

**Figure 2 antibiotics-11-00882-f002:**
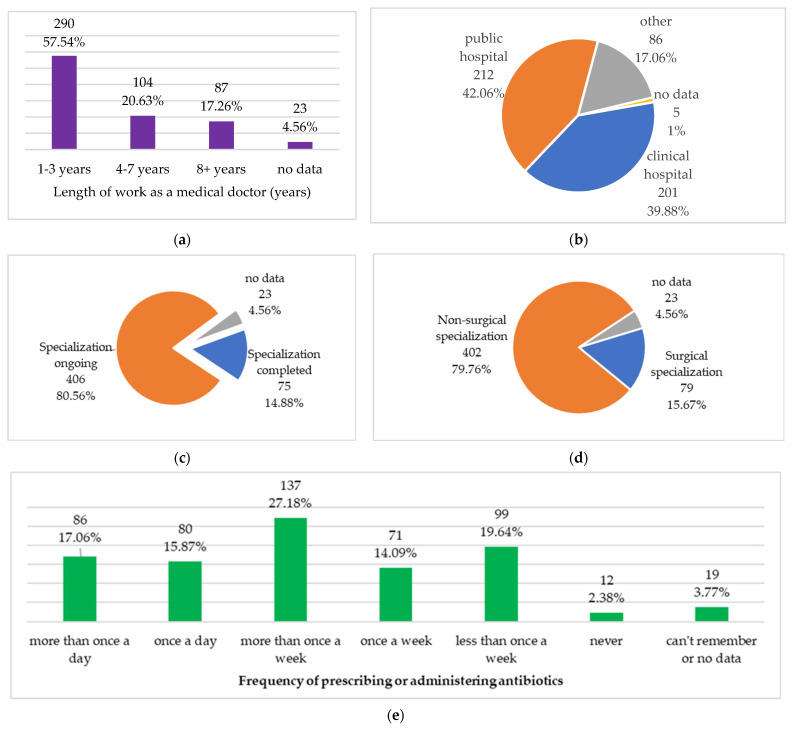
Professional characteristics of the study group: (**a**) length of work as a medical doctor; (**b**) main workplace; (**c**) specialization status; (**d**) type of specialization; (**e**) frequency of prescribing or administering antibiotics.

**Figure 3 antibiotics-11-00882-f003:**
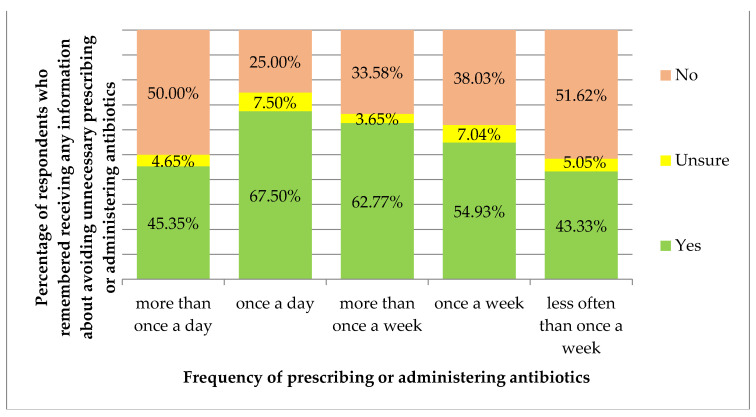
Percentage of respondents who remembered receiving any information about avoiding unnecessary prescribing or administering or dispensing of antibiotics, by the frequency of prescribing or administering antibiotics.

**Figure 4 antibiotics-11-00882-f004:**
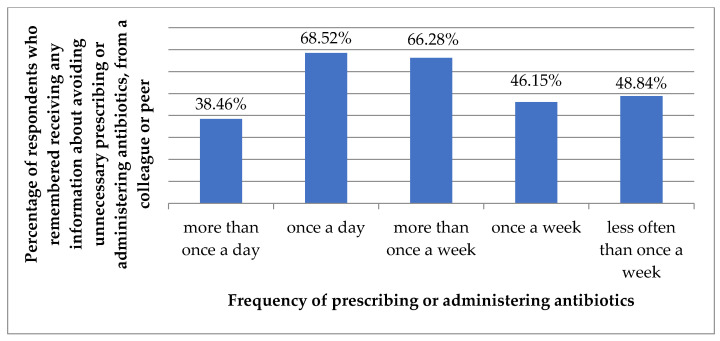
Percentage of respondents who remembered receiving information about avoiding unnecessary prescribing or administering or dispensing of antibiotics from a colleague or peer, by the frequency of prescribing or administering antibiotics.

**Figure 5 antibiotics-11-00882-f005:**
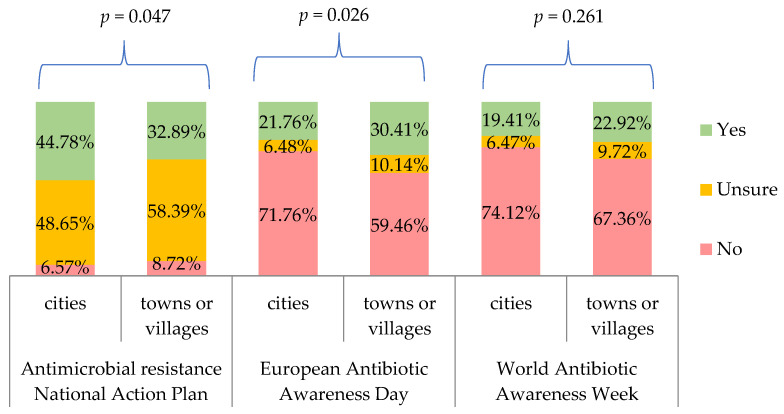
Percentage of respondents who were aware/unaware of whether their country had a national action plan on antimicrobial resistance, who had/had not heard of European Antibiotic Awareness Day, and who had/had not heard of World Antibiotic Awareness Week, by respondents’ place of residence. *p* for chi-square test.

**Table 1 antibiotics-11-00882-t001:** Resources most frequently used by medical doctors in the management of infections (*n* = 504).

Resource	*n* (%)
clinical practice guidelines	451 (89.48)
previous clinical experience	369 (73.21)
continuing education training courses	226 (44.84)
infection specialists	184 (36.51)
scientific journals	167 (33.13)
industry materials/publications	101 (20.04)
documentation from pharmaceutical companies	45 (8.93)
representatives from pharmaceutical companies	9 (1.79)
social media	8 (1.59)
none of the above/other	23 (4.56)

Multiple responses were allowed: up to three responses possible.

**Table 2 antibiotics-11-00882-t002:** Respondents who received information on avoiding unnecessary prescribing, administering or dispensing of antibiotics, and those reporting that the information contributed to changing their views or practice.

Question	Yes	No	Unsure
In the last 12 months, do you remember receiving any information about avoiding unnecessary prescribing or administering or dispensing of antibiotics? (*n* = 504)	270 (53.57)	206 (40.87)	28 (5.56)
Did the information contribute to changing your views about avoiding unnecessary prescribing or administering or dispensing of antibiotics? (*n* = 270)	150 (55.56)	68 (25.19)	52 (19.26)
On the basis of the information you received, have you changed your practice on prescribing or administering or dispensing antibiotics? (*n* = 270)	113 (41.85)	112 (41.48)	45 (16.67)

Results are presented as *n* (%).

**Table 3 antibiotics-11-00882-t003:** Sources of information about avoiding unnecessary prescribing/dispensing/administering of antibiotics and sources of information with the most influence on changing the respondents’ views.

Source of Information	Sources of Information about Avoiding Unnecessary Prescribing/Dispensing/Administering of Antibiotics (*n* = 270) ^1^	Sources of Information Which Had the Most Influence on Changing the Respondent’s Views (*n* = 150) ^2^
Published guidelines	181 (67.04)	102 (68.00)
Colleague or peer	153 (56.67)	37 (24.67)
Training—conference group	148 (54.81)	58 (38.67)
Employer	75 (27.78)	16 (10.67)
Training—individual/specialized	62 (22.96)	26 (17.33)
Newspaper	42 (15.56)	6 (4.00)
Scientific organization	33 (12.22)	10 (6.67)
Social media	31 (11.48)	0 (0.00)
Media advertising (TV/radio)	20 (7.41)	0 (0.00)
Audit and feedback	17 (6.30)	7 (4.67)
Professional body (e.g., doctors/pharmacists/nurses)	12 (4.44)	2 (1.33)
Public policy	7 (2.59)	3 (2.00)
Other	9 (3.33)	4 (2.67)

^1^ Any number of responses allowed. ^2^ Multiple responses were allowed: up to two responses possible. Results are presented as *n* (%).

**Table 4 antibiotics-11-00882-t004:** At what level is raising the issue of antibiotic resistance the most effective? (*n* = 504).

Level	*n* (%)
Action is needed at all levels	299 (59.33)
Individual (by medical doctors prescribing drugs)	141 (27.98)
Individual (by all healthcare workers)	116 (23.02)
Regional/National	55 (10.91)
EU/Global	43 (8.53)
Related to environment/animal health	36 (7.14)
Individual (in public)	18 (3.57)
I don’t know	8 (1.59)

Multiple responses were allowed: up to two responses possible.

**Table 5 antibiotics-11-00882-t005:** Initiatives undertaken in Poland focusing on antibiotic awareness and antibiotic resistance (*n* = 504).

Initiatives	*n* (%)
National or regional guidelines on infection control	286 (56.75)
Conferences/events focused on tackling antibiotic resistance	207 (41.07)
Toolkits of educational materials, including on-line and regional guidelines for healthcare workers	191 (37.90)
National or regional posters or leaflets on antibiotic awareness	165 (32.74)
Articles on antibiotic resistance in the (national) press	132 (26.19)
Awareness raising by professional organizations	115 (22.82)
World Antibiotic Awareness Week (WAAW)/European Antibiotic Awareness Day (EAAD)	86 (17.06)
Television and radio advertising for the public	83 (16.47)
National campaigns	60 (11.90)
I don’t know of any initiatives	63 (12.50)
Other	6 (1.19)

Any number of responses allowed.

**Table 6 antibiotics-11-00882-t006:** Respondents who agreed/disagreed that there had been good promotion of prudent antibiotic use and information about antibiotic resistance in their country, and those who believed the national campaign had been effective in reducing unnecessary antibiotic use and controlling antibiotic resistance in their country.

Answer	There Had Been Good Promotion of Prudent Antibiotic Use and Information about Antibiotic Resistance in Their Country (*n* = 494 Responses Given)	The National Campaign Had Been Effective in Reducing Unnecessary Antibiotic Use and Controlling Antibiotic Resistance in Their Country (*n* = 489 Responses Given)
strongly disagree (1)	136 (27.53)	99 (20.25)
disagree (2)	233 (47.17)	199 (40.70)
undecided (3)	68 (13.77)	91 (18.61)
agree (4)	49 (9.92)	85 (17.38)
strongly agree (5)	8 (1.62)	15 (3.07)

Results are presented as *n* (%).

**Table 7 antibiotics-11-00882-t007:** Respondents who were aware/unaware of whether their country had a national action plan on antimicrobial resistance, who had/had not heard of European Antibiotic Awareness Day, and who had/had not heard of World Antibiotic Awareness Week (*n* = 504).

Question	Yes	No	Unsure
Antimicrobial Resistance National Action Plan	202 (40.08)	35 (6.94)	267 (52.98)
European Antibiotic Awareness Day	120 (23.81)	336 (66.67)	48 (9.52)
World Antibiotic Awareness Week	100 (19.84)	353 (70.04)	51 (10.12)

Results are presented as *n* (%).

**Table 8 antibiotics-11-00882-t008:** Respondents who believe European Antibiotic Awareness Day and World Antibiotic Awareness Week have been effective/ineffective in raising awareness about prudent use of antibiotics and antibiotic resistance in their country (*n* = 504).

Answer	European Antibiotic Awareness Day (*n* = 441 Responses Given)	World Antibiotic Awareness Week (*n* = 440 Responses Given)
very ineffective (1)	47 (10.66)	45 (10.23)
ineffective (2)	69 (15.65)	69 (15.68)
undecided (3)	306 (69.39)	303 (68.86)
effective (4)	17 (3.85)	21 (4.77)
very effective (5)	2 (0.45)	2 (0.45)

Results are presented as *n* (%).

## Data Availability

Database used in this study is available upon request from Wojciech Stefan Zgliczyński. The data are not publicly available due to privacy restrictions.

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
