# Peer review of "Polish Medical Doctors’ Opinions on Available Resources and Information Campaigns concerning Antibiotics and Antibiotic Resistance, a Cross-Sectional Study"

_antibiotics, 2022, doi:10.3390/antibiotics11070882_

Round 1

Reviewer 1 Report

Dear Authors,

The paper addresses a very stringent problem in current publoc health: antibiotic resistance and practical, efficient measures that could deal with this problem.

The study presents data from a significant number of subjects (504), altough they  are mostly from higher level medical institutions (clinical and public hospitals - 413), clinical specialties (406 vs 75 surgical) and mainly residents (406). This bias should be underlined as I think is relevant that the respondents have a limited experience (which is high in their responses) and a shorter, rushed contact with the patients (compared with general practitioners and outpatient clinics).

The study is thourough and attentive to all angles with the inherent limitations of the method (questionnaires – for example in table 8 is unacceptable to have more unsure answers to a yes or no question - Antimicrobial resistance National Action Plan, 267 uncertain vs 237 with a clear answer), which I think have to be underlined in the limitations section of the paper.

In terms of statistics I would ask the authors to mention how was „p” calculated (Lines 142-144).

Author Response

Dear Editors and Reviewers,

Thank you sincerely for all the provided comments and suggestions regarding the manuscript draft. We have amended the paper in compliance with the vast majority of your remarks as they certainly improve the quality of how our study is presented. Detailed explanation and more information has been introduced in the current round of revisions.

We look forward to hearing from you soon.

Authors

------------------------

Response to Reviewer 1 Comments

The authors would like to thank the Reviewer for the comments that helped to improve the quality of the paper. We have carefully analysed all comments and modified the manuscript. Our responses are listed below.

Authors

POINT #1.1
The study presents data from a significant number of subjects (504), altough they  are mostly from higher level medical institutions (clinical and public hospitals - 413), clinical specialties (406 vs 75 surgical) and mainly residents (406). This bias should be underlined as I think is relevant that the respondents have a limited experience (which is high in their responses) and a shorter, rushed contact with the patients (compared with general practitioners and outpatient clinics).

RESPONSE #1.1

Thank you very much for your comment. The characteristics of the study group, including place of work, is briefly described in the article (‘2. Results’). We did not conduct a comparison between our study group and general practitioners working in outpatient clinics. That is why we do not know if there are any differences in their experience. We cannot exclude that in the case of our study group the crucial factor for limited experience was rather relatively young age than a place of work.

POINT #1.2
The study is thourough and attentive to all angles with the inherent limitations of the method (questionnaires – for example in table 8 is unacceptable to have more unsure answers to a yes or no question - Antimicrobial resistance National Action Plan, 267 uncertain vs 237 with a clear answer), which I think have to be underlined in the limitations section of the paper.

RESPONSE #1.2

Thank you for pointing out this variability. We have made an explicit mention in the ‘Discussion’ section of the respondents' indecisiveness in case of this question as we consider it another important finding of the paper - the majority (53%) of doctors in Poland are uncertain about whether the National Action Plan on AMR is in place or not. In the manuscript we also underlined that in the ECDC study, the number of ‘uncertain’ answers from Poland was even higher than in our study (ca. 60%).

This point has also been raised in the newly added subsection of ‘Conclusions’ which was made upon suggestion from another Reviewer (5.1 Recommendations and future directions).

POINT #1.3
In terms of statistics I would ask the authors to mention how was „p” calculated (Lines 142-144).

RESPONSE #1.3
We used chi squared test to compare categorical responses (yes vs no) on questions between men and women.

Reviewer 2 Report

The manuscript under review attempts to evaluate the opinions of Polish medical doctors on available resources and information campaigns on antibiotics and antibiotic resistance in a cross-sectional survey.  In general, the manuscript captured the details of the study design, methodology, and project implementation. All the sections of the manuscript are well written and concluded. The methods and results sections need several changes. Kindly find below the detailed comments and suggestions in all the sections, which will help the authors to check and revise the manuscript.

Title: Kindly rewrite and make it reader easy

Abstract:

1.     Kindly provide a structured abstract

2.     Mention the significant values

3.     Write in brief the conclusion of the study

4.     Provide MeSH keywords

Introduction:

1.     The introduction looks lengthy and diverted a bit from the title and topic

2.     Kindly write briefly about the polish medical practice and antibiotic usage with references

3.     Literature on antibiotic resistance in polish?

4.     Approximately 700,000 people die because of antibiotic resistance in a calendar year; kindly check for polish literature

5.     Refer to and cite the below-mentioned paper

https://www.hindawi.com/journals/bmri/2021/5599724/

Materials and methods:

1.     Kindly shift the M&M before results and after the introduction

2.     Did the authors perform sample size calculations? Provide details

3.     Although participation in the study was anonymous and voluntary, each surveyed participant gave us their verbal consent: how did the authors obtain this?

4.     Mention the ethical statement in section 2.1 as ethical consideration at the start of the M&M section

5.     The questionnaire was delivered to the respondents in paper form: how did the authors deliver it exactly? Time taken? Provide details like no of questions, how and who distributed and collected? Was it in university? Did the authors take approval from the dean or the campus director?

6.     Was the data entered in the sheet? How did they transfer to software?

7.     Statistical test: why did not the authors use the Exact test?

Results:

1.     Table 1 is the repetition of previously published paper of authors; kindly change it to pie charts / Bar graphs

2.     Avoid repetition of table 1 content as text, lines 114 to 129

3.     2.2. Resources used by medical doctors in the management of infections: exact repetition as Table 1 and the text after table

4.     Mention only the significance values in the text; the rest can be seen in the tables

5.     Same for Table 1 to 9 and Figures 1 to 3

Discussion:

1.     Kindly refer to the paper mentioned below

https://www.exeley.com/polish_journal_of_microbiology/pdf/10.5604/01.3001.0010.4856

2.     The authors have written the discussion very well and mentioned the limitations as well

3.     Write in brief about recommendations and future directions

Conclusion:

1.     This study presents the results of a survey performed on a sample of Polish medical doctors which was based on the ECDC study on healthcare workers' knowledge, attitudes and behaviours with respect to antibiotics, antibiotic use and antibiotic resistance. Prescribers in our group rely on resources such as clinical practice guidelines, their own clinical experience, organized trainings and other clinicians' knowledge for navigating antibiotic treatment: kindly delete this

2.     The conclusion should be on the outcome of the study; mention what authors conclude after conducting the current study

   Informed Consent Statement: Informed consent was obtained from all subjects involved in the study upon filling out the questionnaire: provide details

References:

1.     Page numbers are missing in several references

2.     Maintain uniformity

Author Response

Response to Reviewer 2

The authors would like to thank the Reviewer for the comments that helped to improve the quality of the paper. We have carefully analysed all comments and modified the manuscript. Our responses to each point raised by the Reviewer are listed below.

Authors

POINT #2.1

Title: Kindly rewrite and make it reader easy

RESPONSE #2.1.

The title has been rewritten.

POINT #2.2

Abstract:

  1. Kindly provide a structured abstract
  2. Mention the significant values
  3. Write in brief the conclusion of the study
  4. Provide MeSH keywords

RESPONSE #2.2

  1. The abstract was given a structure upon our revision.
  2. The numbers presented in the abstract were not used in any variability tests - they are the overall percentage of respondents who declared one answer over another. Significant values (p-values) are described in detail in the main text (‘2. Results’) where different categories are compared, under each question section. We realised that the word limit on abstract (max 200 words) was too short for presenting detailed results with significant values thus we opted for presenting general percentages only.
  3. Conclusion of the abstract has been modified.
  4. Keywords have been all checked and modified to match the MeSH listing

POINT #2.3

Introduction:

  1. The introduction looks lengthy and diverted a bit from the title and topic
  2. Kindly write briefly about the polish medical practice and antibiotic usage with references
  3. Literature on antibiotic resistance in polish?
  4. Approximately 700,000 people die because of antibiotic resistance in a calendar year; kindly check for polish literature
  5. Refer to and cite the below-mentioned paper

https://www.hindawi.com/journals/bmri/2021/5599724/

RESPONSE #2.3

  1. ‘1. Introduction’ has been shortened - excerpts not supported by our results have been removed.
  2. A paragraph was added on the Polish medical practice and antibiotic use citing new resources.
  3. Polish literature on antibiotic resistance has been included.
  4. The most recent study we found is from 2015. Quoted numbers for Poland have been referenced in the newly introduced paragraph.
  5. Thank you for suggesting this resource. We have included the above-mentioned article in our references list and cited suitably for the context (‘1. Introduction’).

POINT #2.4

Materials and methods:

  1. Kindly shift the M&M before results and after the introduction
  2. Did the authors perform sample size calculations? Provide details
  3. Although participation in the study was anonymous and voluntary, each surveyed participant gave us their verbal consent: how did the authors obtain this?
  4. Mention the ethical statement in section 2.1 as ethical consideration at the start of the M&M section
  5. The questionnaire was delivered to the respondents in paper form: how did the authors deliver it exactly? Time taken? Provide details like no of questions, how and who distributed and collected? Was it in university? Did the authors take approval from the dean or the campus director?
  6. Was the data entered in the sheet? How did they transfer to software?
  7. Statistical test: why did not the authors use the Exact test?

RESPONSE #2.4

  1. The order of the text and the arrangement of particular sections was prepared according to the MDPI template, with ‘Materials and Methods’ section between ‘Discussion’ and ‘Conclusions’.
  2. A total of 504 questionnaires that were correctly completed were included in the study. The study group consisted of 504 doctors participating in specialization courses at the Center for Postgraduate Medical Education in Warsaw (response rate 81%). Due to the size and the compulsory character of the course, it can be considered representative of the group of doctors during specialization training. That is why the authors decided not to perform sample size calculation.
  3. The questionnaire was delivered to the respondents in paper form by the researcher at the beginning of every course conducted by CMKP. The researcher informed every group about the survey purpose and details. The students could decide whether to take part in the survey or not. Modifications were introduced in the ‘4. Materials and Methods’ section of the manuscript to reflect upon this process (--> 4.2.2 Dissemination process and data collection’)
  4. The ethical statement was shifted into a separate subsection ‘4.4. Ethical statement’ in the ‘4. Materials and Methods’ section.
  5. A self-administrated questionnaire was prepared in Polish on the basis of the validated tool used by the ECDC. The tool was described in a technical report and the sources are provided in the ‘References’ section. Study was conducted with full approval of the Dean of the School of Public Health, Centre for Postgraduate Medical Education in Warsaw, Poland (-> ‘4.4. Ethical statement). Detailed and extended description of the questionnaire dissemination and collection process was provided in the ‘4. Materials and Methods’ subsection ‘4.2. Survey questionnaire’ (--> 4.2.2 Dissemination process and data collection’).
  6. Data was collected in paper form. All obtained questionnaires were transferred to the electronic database by the office workers of the School of Public Health CMKP. The correctness of data transfer was verified on an ongoing basis by a researcher and co-author of the paper. A modification was introduced into the ‘4.2. Survey questionnaire’ subsection of the ‘Materials and methods’ section of the manuscript.
  7. We used Pearson's chi-square test or Fisher’s exact test to correlate two categorical variables. The latter test was used only if at least expected count was smaller than 5. We added this information into Statistical methods.

POINT #2.5

     Results:

  1. Table 1 is the repetition of previously published paper of authors; kindly change it to pie charts / Bar graphs
  2. Avoid repetition of table 1 content as text, lines 114 to 129
  3. 2.2. Resources used by medical doctors in the management of infections: exact repetition as Table 1 and the text after table
  4. Mention only the significance values in the text; the rest can be seen in the tables
  5. Same for Table 1 to 9 and Figures 1 to 3

RESPONSE #2.5

  1. We deleted table 1 and changed it for pie charts and bar graphs (Figure 1 and Figure 2).
  2. We deleted those lines.
  3. We deleted the repetitions in the text.
  4. We corrected it accordingly.
  5. We deleted the repetitions in the text.

POINT #2.6

Discussion:

  1. Kindly refer to the paper mentioned below

https://www.exeley.com/polish_journal_of_microbiology/pdf/10.5604/01.3001.0010.4856

  1. The authors have written the discussion very well and mentioned the limitations as well
  2. Write in brief about recommendations and future directions

RESPONSE #2.6

  1. We regret to inform you that this resource is not available anymore or has been accessed via internal network, or the address has been misspelt thus we could not cover it within our manuscript.
  2. Thank you very much for your generous words.
  3. A paragraph about recommendations and future directions has been incorporated within the ‘5. Conclusion’ section.

POINT #2.7

Conclusion:

  1. This study presents the results of a survey performed on a sample of Polish medical doctors which was based on the ECDC study on healthcare workers' knowledge, attitudes and behaviours with respect to antibiotics, antibiotic use and antibiotic resistance. Prescribers in our group rely on resources such as clinical practice guidelines, their own clinical experience, organized trainings and other clinicians' knowledge for navigating antibiotic treatment: kindly delete this
  2. The conclusion should be on the outcome of the study; mention what authors conclude after conducting the current study

RESPONSE #2.7

  1. The excerpt indicated has been deleted.
  2. The ‘5. Conclusions’ section has been remodelled and amended in the newly provided version of the manuscript.

POINT #2.8

   Informed Consent Statement: Informed consent was obtained from all subjects involved in the study upon filling out the questionnaire: provide details

RESPONSE #2.8

Modifications were introduced in ‘4. Materials and Methods’ section of the manuscript to reflect upon this process (--> 4.2.2 Dissemination process and data collection’ subsection)

POINT #2.9

References:

  1. Page numbers are missing in several references
  2. Maintain uniformity

RESPONSE #2.9

Thank you for pointing out the mistakes in the 'References' section. They have been corrected appropriately.

Round 2

Reviewer 2 Report

Dear Authors,

The manuscript has much improved and looks presentable in its current form. The only concern left unanswered was that sample size calculation has not been performed; can the authors explain the required numbers of participants that have been included in the study to justify the size?

All the very best 

Best regards

Author Response

Reviever 2

The manuscript has much improved and looks presentable in its current form. The only concern left unanswered was that sample size calculation has not been performed; can the authors explain the required numbers of participants that have been included in the study to justify the size?

Response to Reviewer 2

The authors would like to thank the Reviewer for the comment concerning the sample size calculation. Below we provide explanation of the method used.

According to the data of the Polish Chamber of Physicians and Dentists there are 156 816 medical doctors as of 01/06/2022 (1). Using two sample size calculators (2, 3) we obtained a sample size of 384 assuming 95% confidence level, margin of error 5%, population proportion 50% and population size 156816. Our sample size used was N=504 thus higher than recommended. If you consider this information belongs in the manuscript, we would make the necessary amendment.  

The participation in the courses conducted by the School of Public Health, Center of Postgraduate Medical Education (CMKP) is compulsory for each of around 25 000 physicians currently undertaking specialty in Poland. The participants (respondents) were selected randomly and represented different regions and healthcare institutions from all over Poland. We approached physicians attending the training courses held from October 2019 to March 2020. All 622 physicians attending these courses were eligible to take the survey. A total of 504 questionnaires that were correctly completed were included in the study, which means the response rate was 81%.

Authors

(1) https://nil.org.pl/uploaded_files/1654088370_za-maj-2022-zestawienie-nr-01.pdf

(2) http://www.raosoft.com/samplesize.html

(3) https://www.calculator.net/sample-size-calculator.html?type=1&cl=95&ci=5&pp=50&ps=156816&x=90&y=7